# Quality of Information on Medication Abortion in Private Pharmacies: Results from a Mystery Client Study in Kinshasa, Democratic Republic of Congo

**DOI:** 10.3390/healthcare13050491

**Published:** 2025-02-24

**Authors:** Denise P. Ngondo, Pierre Z. Akilimali, Nguyen Toan Tran, Nadia Lobo, Dynah M. Kayembe, Francis K. Kabasubabo, Mike Mpoyi, Jean-Claude Mulunda, Grace Sheehy, Paul Samson Dikassa Lusamba

**Affiliations:** 1Patrick Kayembe Research Center, Kinshasa School of Public Health, University of Kinshasa, Kinshasa P.O. Box 11850, Democratic Republic of the Congo; denise.ngondo@unikin.ac.cd (D.P.N.); dirchkayembe@yahoo.fr (D.M.K.); fkabasu13@gmail.com (F.K.K.); paul.lusamba@unikin.ac.cd (P.S.D.L.); 2Department of Epidemiology and Biostatistics, School of Public Health, University of Kinshasa, Kinshasa P.O. Box 11850, Democratic Republic of the Congo; 3Department of Nutrition, School of Public Health, University of Kinshasa, Kinshasa P.O. Box 11850, Democratic Republic of the Congo; 4Australian Centre for Public and Population Health Research, Faculty of Health, University of Technology Sydney, P.O. Box 123, Sydney, NSW 2007, Australia; nguyentoan.tran@uts.edu.au; 5Faculty of Medicine, University of Geneva, Rue Michel-Servet 1, 1206 Geneva, Switzerland; 6Ipas Democratic Republic of Congo (DRC), Kinshasa P.O. Box 1213, Democratic Republic of the Congo; jivenl@ipas.org (N.L.); mpoyim@ipas.org (M.M.); mulundajc@ipas.org (J.-C.M.); 7Ipas, P.O. Box 9990, Chapel Hill, NC 27515, USA; sheehyg@ipas.org

**Keywords:** medical abortion, pharmacies, misoprostol, mystery client, information, counseling, quality, providers

## Abstract

**Introduction**: Pharmacies are important points of access and information for women seeking medication abortion. In the context of the Democratic Republic of Congo [DRC], where the legal conditions for abortion have expanded in recent years and now allow pharmacies to dispense medication abortion with a prescription, little is known about medication abortion counseling and care offered by pharmacy staff. The aim of this study was to explore the quality of information provided by pharmacy staff to customers seeking medication abortion in Kinshasa. **Methodology**: A cross-sectional study using the mystery client (MC) approach was conducted in 480 pharmacies between April and May 2023. Trained female (*n* = 9) and male (*n* = 3) investigators played the role of mystery clients seeking abortion medication for themselves (or their partner or relative), and they asked questions to assess the performance of pharmacy staff and the quality of the provided information. The MCs recorded the results of their visits immediately after they left the pharmacy. Data were analyzed using Stata 17.0 and QGIS. The research protocol received ethical approval from the Kinshasa School of Public Health, and the need for informed consent was waived as pharmacy providers were being observed acting in their professional capacity. **Results**: Misoprostol was available at 40% of pharmacies visited, while mifepristone–misoprostol was available at less than 2%. Correct dosage information for misoprostol was provided by only 23% of the providers, with the lowest proportion observed in interactions involving male partners (2.4%). During discussions, only 10.6% of the providers explained what to expect during the abortion process. The quality of information differed according to the client profile, the district, and whether the client had a prescription. **Conclusions**: While medication abortion can provide a safe option for women seeking to manage their own abortion, the lack of adequate information on the correct dosage and administration can hinder the effectiveness of this regimen. To fully realize the potential of this medication for reducing unsafe abortion, it is essential that pharmacy staff are trained and supported to provide high-quality information and services, and that inequities in access to medications are addressed.

## 1. Introduction

Unsafe abortion remains a major public health challenge worldwide. According to Bearak et al. [1], between 2015 and 2019, 61% of unwanted pregnancies ended in abortion, representing an overall rate of 39 abortions per 1000 women aged 15 to 49. Nearly half of these abortions are carried out in unsafe conditions, with serious consequences for women’s health, including severe complications and avoidable deaths [2,3].

In sub-Saharan Africa, the number of abortions almost doubled between 1995 and 1999 and between 2015 and 2019, rising from 4.3 million to 8.0 million, reflecting an increase in demographic growth in the region [4]. In the Democratic Republic of Congo (DRC), the maternal mortality rate remains one of the highest in the world, at 846 deaths per 100,000 live births [5]. Unsafe abortion is one of the main causes of death among women of childbearing age [6]. However, in 2018, abortion became legal in the DRC due to the country’s ratification of the Maputo Protocol, and in line with the indications described in the Protocol: sexual assault, rape, incest, and in cases where the pregnancy endangers the life or physical or mental health of the woman [7]. The Ministry of Health has also authorized use of mifepristone–misoprostol for medication abortion, added mifepristone to the essential medicines list, and expanded the cadres of providers that can offer abortion care [7].

In Kinshasa, recent studies have revealed a rate of 105 abortions per 1000 women, well above the regional average of 34 per 1000 women [8]. The World Health Organization [9] recommends access to safe abortion services at all levels of the healthcare system. In contexts where the laws on abortion have recently been liberalized, such as in the DRC, women often turn to community informants and pharmacies to obtain abortifacients or abortion-related information [10,11,12]. When following proper protocols, pharmacy providers (including both pharmacists and salespeople) play an essential role in ensuring access to medication abortion, which can be safely used outside formal clinical healthcare settings when taken according to WHO guidelines [13,14].

Pharmacies are essential in delivering abortion services for various reasons. Firstly, they are firmly entrenched in the community, rendering them accessible to those in need. Secondly, they function as key sources of health information broadly and about abortion specifically. Ultimately, pharmacies assist consumers in maneuvering through the stigma, bureaucracy, and exorbitant expenditures associated with the healthcare system. In a constrained environment like the DRC, women depend on dubious information sources within their communities and utilize unverified products, including therapeutic plants and several others. The pharmacy’s project aims to replicate this established community practice by altering the source and quality of information and products. This method provides the woman with privacy and secrecy, mitigating the risk of stigma. Advancements in technology and enhanced access to mobile Internet, particularly in remote regions, have led to a growing availability of abortion information via applications. Access to goods via pharmacists frequently influences the realization of a woman’s choices [15]. However, several studies indicate that pharmacy providers often lack appropriate knowledge about medication abortion protocols and client counseling [14,16]. For example, research in Bogotá, India, and Nigeria have all found that clients seeking medication abortion often received inadequate or incorrect guidance on the correct use of the medications [17,18]. “Existing literature reports several studies using the mystery client methodology to assess access to and the dispensing of abortion medications in similar contexts. A study conducted in Nigeria examined the knowledge and practices of drug sellers in dispensing misoprostol for abortion. It found that 86% of clients received the appropriate dosage, and the majority indicated that they would return to the proprietary patent medicine vendor for future services [19]. Additionally, another study conducted in Nigeria explored women’s self-reported experiences using misoprostol obtained from these vendors. Findings showed that 94% of participants reported complete abortion without surgical intervention, although only a few sought cares at a health facility [13]. In India, a study conducted in Uttar Pradesh used mystery clients to assess whether pharmacists provided different information based on clients’ gender and marital status. Results indicated that pharmacists were less likely to provide comprehensive information to female clients, particularly unmarried women [20]. Another study in Nepal examined the dispensing practices of private pharmacy workers using mystery clients. It found that 70% of pharmacy workers inquired about the date of the last menstrual period, and 66.4% provided information on the route of drug administration. However, only 55.9% informed clients about potential side effects [21]. Additionally, 35.7% of visits resulted in the dispensing of medication abortion drugs, while 39.3% were refused, primarily due to the lack of a prescription and referrals to other health facilities [21]”.

This knowledge gap could lead to unsafe and ineffective practices.

There are limited studies in the literature on the quality of counseling on medication abortion by pharmacy providers in the DRC. This research aims to fill the knowledge gap by assessing pharmacy providers’ knowledge, counseling, and dispensing practices in Kinshasa using a mystery client methodology. Further, the use of this methodology offers a unique opportunity to assess the quality-of-service delivery, particularly for a stigmatized service such as abortion care, which is often underreported. This research contributes to the small but growing body of literature that attempts to document medication abortion dispensing practices in pharmacies in low- and middle-income countries.

## 2. Methodology

### 2.1. Study Design, Study Population, and Sampling

A cross-sectional mystery client study was conducted in Kinshasa pharmacies 1 February–2 March 2023 as part of IPAS’ evaluation of the country’s progress under the Maputo Protocol, especially in the application of sexual and reproductive health rights [22]. IPAS and its partners are supporting the Ministry of Public Health by training service providers, particularly those working in pharmacies, to increase their level of knowledge and improve the quality of services provided to women seeking information on medical abortion. Before the data collection began, members of the research team identified and confirmed pharmacy addresses in Kinshasa and confirmed whether they were still in operation based on the updated list of pharmacies in this network, and a simple random sampling technique was used to select 480 pharmacies. The sample size was determined to accurately evaluate the proportion of adequate information provided by pharmacies in Kinshasa regarding abortion medication. A proportion of 50% was used due to the absence of sufficient data throughout the study’s design phase. Utilizing the Kish Leslie formula for power calculation with a 5% margin of error and a 95% confidence interval, the minimum requisite sample size was determined to be 384. We incorporated a 20% contingency for potential issues such as potential inaccuracies in pharmacy addresses, refusal of specific pharmacists to serve clients, or the closure of pharmacies during mystery client visits, resulting in a total of 480 pharmacies being surveyed in this study.

These 480 pharmacies were randomly selected from the 35 health zones of the city of Kinshasa from a list of 1694 pharmacies established in 2021 by the National Reproductive Health Program. Drug shops/pharmacies in the difficult-to-access military health zone (Kokolo) and a very distant rural health zone (Maluku 2) were excluded for convenience reasons. This sampling frame was used during the previous mystery client study [23]. The service provider (which included both pharmacists and salespeople) present at the point of sale on the day of the mystery client’s visit was observed and interviewed.

### 2.2. Geographic Distribution of Participating Pharmacies

The geographic distribution of the private pharmacies surveyed by the mystery clients in Kinshasa is shown in Figure 1. The map also shows the locations of the pharmacies [red dots] in the administrative and geographical regions of the city of Kinshasa, demonstrating urban disparities in the distribution of pharmacy-based services. The pharmacies are highly concentrated in the central urban zones of Gombe, Barumbu, and Lingwala, as well as in relatively high-population areas such as Makala, Kasa-Vubu, and Lemba. Some of the remote zones, such as Mont Ngafula and Nsele, have a lower density of pharmacies that were sampled.

### 2.3. Data Collection

Twelve mystery shoppers, including nine females and three males with public health and family planning survey experience, were trained over 4 days in medical abortion protocols and data collection procedures. They enacted three profiles—(1) a woman seeking misoprostol; (2) a male partner seeking misoprostol for his girlfriend; and (3) a close relative seeking misoprostol for her daughter who wishes to have an abortion. Each sampled pharmacy was visited by only one mystery client. The scripts used during the visits are appended to this article (Appendix A English and French versions).

Three possible scenarios at the pharmacy were envisaged: (1) the client gave the medical prescription spontaneously; (2) the client showed the medical prescription at the request of the provider; and (3) the client did not need to show the medical prescription.

To ensure data consistency, the mystery clients were trained to provide specific answers to the different scenarios and request additional information. The mystery clients were instructed to gather as much information as possible regarding whether the pharmacy staff would be prepared to provide a drug that might terminate the pregnancy and information on related usage instructions, physical effects, and any complications. The mystery clients were also asked to remember the elements of this interaction and the atmosphere surrounding the conversation. At the end of their conversation, the mystery client found a reason to interrupt the interview and leave the pharmacy. To reduce recall bias, once the MC leaves the pharmacy, she (he) will immediately complete a MC visit report on the study smartphones, including the recording of the GPS coordinates of the point of sale for data validation purposes. The tablets were configured with the surveyCTO application. The completed form was uploaded to the server at the end of the working day.

### 2.4. Key Variables and Their Measurements

A 12-point information quality score was used to assess provider communication regarding the following items: the date of the last menstrual period or the month of pregnancy, whether the pregnancy was confirmed, how the pregnancy was confirmed, the reasons for the abortion, the exact instructions for the proposed drug dosage, the correct route of administration, what to expect during the abortion process, at least one of the possible and exact side effects that may be observed during the abortion procedure, complications that may arise during the abortion procedure, where to go in the event of complications, and whether to consult a clinician on the fifteenth day after medication administration to confirm abortion completion. The 12 questions that make up the quality score we used in this article come from a similar study conducted in Nepal and Nigeria [21,24]. We used a cutoff of 75%, which was a modification of the Bloom’s cutoff point [25,26,27,28,29,30].

Each question was scored as 1 if the MC confirmed that the provider performed the action and 0 if the provider did not. Each provider received a total score from 0 to 12, converted to a percentage. The providers were then classified into two groups—inadequate information [scores < 75%] and adequate information [scores ≥ 75%]. Quality scores were only calculated for pharmacies that could actually dispense misoprostol to the mystery clients. Misoprostol costs were documented in USD and grouped into three price ranges, as follows: <USD 2, USD 2–5, and >USD 5.

### 2.5. Data Analysis

Data were processed and analyzed in Stata 17 [StataCorp., College Station, TX, USA], with statistical significance set to *p* < 0.05. Categorical variables related to the information provided by the pharmacists were summarized as relative frequencies. Confidence interval plots visualized differences in adequate information provision and quality scores across districts, scenarios, and prescription status. Box-and-whisker plots displayed misoprostol cost variations across these parameters. Due to non-normal distributions, Kruskal–Wallis tests were used to assess differences in costs and quality scores. The quality of the information in this article was further examined only for the pharmacies that supplied misoprostol to the mystery clients. A quantum geographic information system (QGIS), 3.24.3-Tisler QGIS, was used to map the visited pharmacies.

### 2.6. Ethical Considerations

The research protocol received ethical approval from the Kinshasa School of Public Health (ESP/CE/04/2023). Given the context of DRC, where abortion has long been condemned by the penal code, and considering the Congolese culture where abortion is an almost “taboo” subject, this study would evaluate the knowledge and practices of pharmacy providers on medical abortion, which would be subject to information bias if the providers were informed that they were participating in a study. They would change their behavior, and the study would not achieve its objectives or would have biased results. Consistent with mystery client methodology, provider informed consent was waived with ethics committee approval (ESP/CE/04b/2023), as pharmacy providers were being observed in a public setting acting in their professional capacity [31]. The findings are provided in an aggregated format, obscuring which pharmacy provides specific services. Additionally, we altered the GPS coordinates to prevent the identification of the precise locations of the pharmacies, and we utilized counseling timestamps to prevent the identification of the pharmacy through travel data. Aside from the modified GPS data intended to maintain the pharmacy’s privacy, we have neither gathered nor disseminated any more information that could reveal the pharmacy’s identity.

## 3. Results

Table 1 shows that misoprostol was available in 40% of the 480 pharmacies visited, while both mifepristone and misoprostol were available in less than 2%. In the pharmacies where medication abortion pills were available, the majority of the mystery clients successfully acquired either misoprostol (94%) or mifepristone (5.8%). Looking across the four districts, the Mont Amba district had the fewest pharmacies stocking misoprostol (32%) compared to Lukunga, which had the most (44%). The mystery client scenarios showed distinct distribution patterns across the four districts of Kinshasa. This heterogeneous distribution across districts and scenarios facilitated a comprehensive assessment of pharmacy provider responses to different client profiles.

Table 2 shows the information given by the pharmacy providers based on the MC scenarios. The scenarios included encounters between pharmacy providers and pregnant women (*n* = 270), pregnant women’s mothers (*n* = 90), and male partners (*n* = 120) for a total of 480 interactions. The nature of the information provided by the service provider was also related to the client’s profile. Correct dosage information for misoprostol was provided by only 23.2% of the providers, with the lowest proportion observed in interactions involving male partners (2.4%). During discussions, only 10.6% of the providers explained what to expect during the abortion process. Nearly half of the providers (47.7%) inquired about the reasons for the abortion, while approximately one in three (35.8%) asked whether the pregnancy had been confirmed. The providers were more likely to seek pregnancy confirmation from the male partner than in the other MC scenarios. Only 24.8% of the providers inquired about how the pregnancy had been confirmed, regardless of the scenario. The most commonly provided information was on at least one possible side effect of abortion (59.3%) and correct instructions for the route of administration (47.8%). Only 18.4% of the providers gave correct information on potential complications during the abortion process, with male partners receiving this information more frequently than the other MC groups. About one in four MCs (24.9%)were informed on where to seek care in the case of complications, with this guidance being more common for male partners than for the other MC categories. Mothers received insufficient information, particularly on medical follow-up and where to seek care in the case of complications.

### 3.1. Information Quality on Misoprostol According to District, Mystery Client Scenario, and Whether Medical Prescriptions Shown

Among the pharmacies that reported stocking and selling misoprostol, only 11.34% (22/194) provided adequate information about medical abortion. The analysis, as shown in Figure 2, revealed significant variations in information quality across districts, client scenarios, and prescription presentation [*p* < 0.001 for all comparisons]. A geographic distribution analysis showed that the Lukunga district had the highest proportion of providers offering adequate information, while Funa and Mont Amba districts demonstrated notably lower proportions. Client characteristics significantly influenced information quality. Male partners received more comprehensive information than pregnant women, while the mothers of pregnant women received the least adequate information. Counterintuitively, when prescriptions were presented spontaneously during visits, the providers delivered lower-quality information than during visits without prescription presentation.

### 3.2. Cost Analysis of Misoprostol Availability and Distribution

Overall, the median cost of misoprostol was 3.15 USD. Figure 3 shows the median costs of misoprostol, which were comparable across all districts, suggesting general price homogeneity: Funa (3.45 USD), Lukunga (3.26 USD), Mont Amba (3.48 USD), and Tshangu (2.61 USD). However, Funa and Mont Amba districts exhibited a greater price dispersion, with costs ranging up to USD 20–25, indicating that some patients face substantially higher acquisition costs. In contrast, the Tshangu district demonstrated minimal price dispersion and fewer outliers, reflecting more standardized pricing.

A cost analysis using CM profiles revealed consistent median prices across pregnant women, their mothers, and male partners, suggesting that the purchaser profile does not significantly influence pricing. The limited price dispersion across these categories further supported pricing uniformity. A prescription presentation analysis showed comparable median costs when prescriptions were displayed. However, scenarios without prescription presentation, while maintaining similar medians, showed increased price variation, with values exceeding USD 20, suggesting a potential price elevation in these circumstances.

A statistical analysis using Kruskal–Wallis tests confirmed these observations, revealing no significant differences in misoprostol costs across groups. Specifically, cost distributions were statistically similar across districts (*p* = 0.2768), MC scenarios (*p* = 0.8396), and prescription presentation conditions (*p* = 0.1697). While median prices remained relatively uniform across all studied variables, notable outliers in certain districts indicated localized pricing disparities.

## 4. Discussion

This study aimed to assess the quality of the information provided by pharmacy providers, including qualified pharmacists and salespeople to clients seeking medication abortion in Kinshasa. The results show not only that the quality of the information provided was generally inadequate with notable disparities based on geographical and socio-demographic contexts, but also an extremely limited availability of the combined mifepristone–misoprostol medication for medical abortion in pharmacies in Kinshasa.

While the majority of MCs who obtained misoprostol were informed about at least one potential side effect, information on the correct dosage, potential complications, and where to go for further care in case of complications was infrequently provided. This deficiency was particularly pronounced for certain mystery client profiles, such as mothers. The lack of comprehensive information provided to clients is concerning, as it could increase the likelihood of incomplete abortion requiring further treatment, as well as fear and anxiety among clients.

A growing body of research has assessed the availability and quality of medication abortion dispensing in a range of settings including Kenya, India, Nigeria, and Senegal, using both mystery clients as well as direct interviews with pharmacists [14,18,32,33]. For instance, similar to our findings, a mystery client study in Nigeria found that 51% of facilities surveyed offered medication abortion pills, primarily the misoprostol-only regimen, and only 26% provided correct instructions on administration [18]. In face-to-face interviews with pharmacists in Senegal, just 35% reported selling misoprostol, and rarely for reproductive health indications [14] The incomplete provision of information by pharmacy staff found in this study aligns with the existing literature, which largely has found pharmacy staff knowledge and counseling on abortion pills to be inadequate, and often inaccurate [20,21,32,33,34]. In Nepal, Sigdel et al. [21] found that only 38.5% of pharmacy workers provided information about recommended dosage, while merely 35% discussed potential complications. Similarly, research in Kenya by Reiss et al. [32] reported that just 19% of pharmacy workers demonstrated adequate knowledge of medical abortion medications. For example, mystery client data from 92 pharmacies and chemist shops in Nigeria found that just 26% provided correct information on the proper administration of abortion pills [18]. One possible reason for this could be the lack of training of some providers on the subject. To remedy this shortcoming, government and organizations working to promote safe abortion care could set up targeted training programs. These initiatives would help improve pharmacy providers’ skills and ensure the provision of comprehensive information, an essential condition for ensuring the safe and effective use of medication abortion.

Our findings echo research from Mexico by Lara et al. [34], who found significant variations in information provision based on client characteristics, with only 28% of pharmacy workers providing complete dosing information. Similar patterns emerge from studies in Nepal [21] and India [33], suggesting this may reflect a broader regional challenge in pharmacy-based abortion care. Another study in India highlighted the generally low quality of information provided by pharmacists, particularly concerning the correct timing and dosage of misoprostol, as well as potential side effects [18].

These information gaps appear particularly pronounced for certain client profiles, notably mothers. Male partners received better-quality information than the other groups although very few received accurate dosage information. This reflects a tendency for providers to adapt their communication according to the type of client and demonstrates the important role men often play in procuring abortion pills for their partners. Gender sensitivities and roles within the community may explain this difference. Due to stigmatization, women may feel uncomfortable staying in a pharmacy for an extended period, fearing suspicion. In contrast, when a man is involved—regardless of whether the pharmacist is a woman—the amount of time spent in the pharmacy receiving information is less likely to be questioned. While men received less correct information on dosage, they received more information on possible complications than our other client profiles; given the important role men can often play in women’s abortion trajectories, this information asymmetry could have important complications for women’s correct use of abortion pills. This could be due to gendered expectations about who will make decisions about seeking follow-up treatment, or to ensure male partners can properly take care of their spouse after the abortion.

Past research in Kinshasa has documented the role male partners often play as intermediaries in seeking out abortion sources [10]. Similarly, research in India found that some pharmacy providers felt uncomfortable explaining to women how to use misoprostol, whereas they felt more comfortable talking to men [33]. Another study found that male mystery clients were more likely to be provided MA pills than female mystery clients [18]. Research in Nigeria and Cote d’Ivoire has also found that partner involvement in abortion trajectories is associated with safer abortion care, highlighting the importance of ensuring men have access to correct and high-quality abortion information, and that pharmacists appropriately counsel all clients regardless of gender [35]. Some providers seemed less inclined to provide detailed information about medical abortion to mystery clients than to their male counterparts, especially when these women presented themselves as unmarried [20]. These behaviors may be influenced by cultural norms or gender stereotypes, which shape the interactions between service providers and their customers and could limit access to necessary abortion information and resources, particularly for young and/or unmarried women. Another reason could be the fear to provide abortion services to young unmarried women because of the persistent criminalization.

With regard to the display of medical prescriptions, the results show that the quality of the information provided was lower when medical prescriptions were displayed. This could indicate that providers assume that customers with a medical prescription are already well informed and require less advice, which would lead to negligence in communicating essential information. If providers consider spontaneous medical prescription display as proof of knowledge on the part of customers, then they may deliberately omit details of useful information to provide to the customer. This might suggest that providers feel obliged to explain the product or treatment further when the prescription is not spontaneously provided. These observations highlight a critical need for standardized communication protocols regardless of sufficient provider explanation and the client understanding the process.

Enhancing the quality of counseling is a significant challenge. Strengthening competencies by providing tailored training and practical checklists and building confidence by creating a supportive legal and social environment may be essential. The improvement relies not only on training but also on various environmental factors, such as limited space, commercial settings that handle multiple clients, cultural gender biases, stigmatization, and ambiguous policies regarding abortion. These factors contribute to the barriers that need to be addressed.

The observed geographic disparities in information quality merit particular attention. A comprehensive study in Uttar Pradesh, India by Diamond-Smith et al. [34] similarly found significant regional variations in provider knowledge and counseling quality, with urban–rural disparities being particularly notable. In our study, the superior performance of the providers in the Lukunga district, contrasted with significantly lower quality scores in Funa and Mont Amba districts, suggests systemic differences that may be attributed to variable provider training in WHO protocols, differences in the socio-economic level of the populations served, and infrastructural resources. These findings highlight the need for targeted interventions in underperforming districts. An effective strategy could include allocating pharmacies to different partners working on the promotion of safe abortion care information and services in order to ensure regular monitoring and improve the quality of the information given to clients.

Given the strides made in expanding access to safe abortion care in the DRC since ratifying the Maputo Protocol, ensuring that pharmacies are well-equipped to provide this essential service is imperative to expanding equitable access to safe abortion care. Pharmacy staff must be equipped with the appropriate information, knowledge, and experience to adequately counsel patients in order to facilitate access to high-quality medication abortion services for all who seek them, regardless of socio-demographic background or geographic location. Standardized training programs for pharmacy staff could improve knowledge and ensure pharmacists are trained to provide accurate, non-judgmental counseling to clients seeking medication abortion. Further, increasing opportunities for abortion-seekers to directly access information on the use of medication abortion, through websites, hotlines, pamphlets, or medication inserts, could further improve equitable access to safe, self-managed abortion.

This study has some limitations. First, there was a possibility of misreporting or recall bias by the mystery clients, although it was minimized through the completion of the structured survey questionnaire immediately after their exit from the pharmacy. While immediate post-visit surveys and experienced interviewer selection mitigated recall bias, some information sequences may have been omitted. Second, the identity of the mystery clients could have been compromised and the pharmacy workers may have behaved differently toward them. Third, we constructed a simple scale to assess the quality of care, assigning equal weight to all the questions in each domain with no external validation of the scale or evidence of its impact on outcomes. However, there are currently no published validated scales to evaluate the quality of care for medication abortion amongst drug stores, and the variables included were chosen based on the WHO guidelines for medication abortion provision to reflect recommended practices during the study. Fourth, the sensitive nature of abortion services and providers’ unfamiliarity with recent liberalization policies may have contributed to underreporting. Fifth, the tripartite mystery client design resulted in small subgroup samples, potentially limiting statistical power. Finally, in our sampling frame, some pharmacies could be missed, which may be more likely to provide abortion services. We lack assurance that every mystery client adhered to the script accurately. The low proportion of pharmacies with misoprostol available reduced the size of the analyses regarding the quality of information provided to clients about medical abortion. We started with an initial sample of 480 pharmacies, and only 194 had misoprostol available. This study was cross-sectional; a future longitudinal study could examine the dynamics of information quality acquired in pharmacies in relation to policy changes.

## 5. Conclusions

To the best of our knowledge, this study is the first to assess the quality of information provided on medical abortion by pharmacy staff and the provision of services through mystery clients in Kinshasa. The survey used random sampling to ensure that the pharmacies included in the study were broadly representative.

This study demonstrates significant deficiencies in pharmacy providers’ medical abortion counseling, with notable variations across geographic districts and client profiles. The particularly low information quality scores observed in Funa and Mont Amba districts, coupled with systematic disparities across client scenarios, indicate critical areas requiring intervention.

Several recommendations emerge from these findings. First, targeted provider training programs should prioritize underperforming districts, particularly Funa and Mont Amba, to address geographic disparities in information quality. Second, the implementation of standardized communication protocols is essential to ensure consistent and comprehensive information delivery, regardless of client profile or prescription status. Third, client education initiatives should emphasize the importance of active information seeking during pharmacy consultations. Fourth, the accessibility of the combined mifepristone–misoprostol formulation for medication abortion in pharmacies across Kinshasa should be enhanced.

Future research directions should investigate the underlying causes of observed geographic and demographic disparities in information provision. Additional studies examining provider attitudes and client perspectives would be valuable to inform the development of targeted interventions in order to enhance medical abortion counseling quality in pharmacy settings in Kinshasa, as well as in other provinces in the country.

## Figures and Tables

**Figure 1 healthcare-13-00491-f001:**
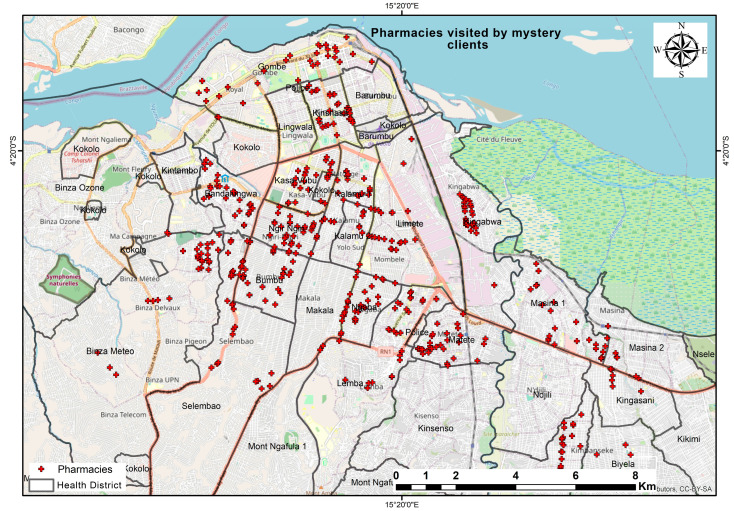
Distribution of private pharmacies visited in the city of Kinshasa.

**Figure 2 healthcare-13-00491-f002:**
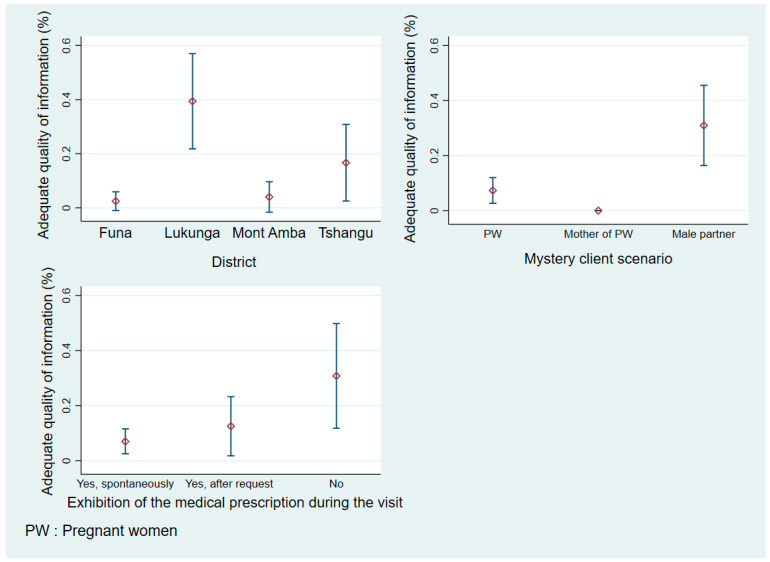
Adequate information provision according to district, mystery client scenario, and whether medical prescriptions shown.

**Figure 3 healthcare-13-00491-f003:**
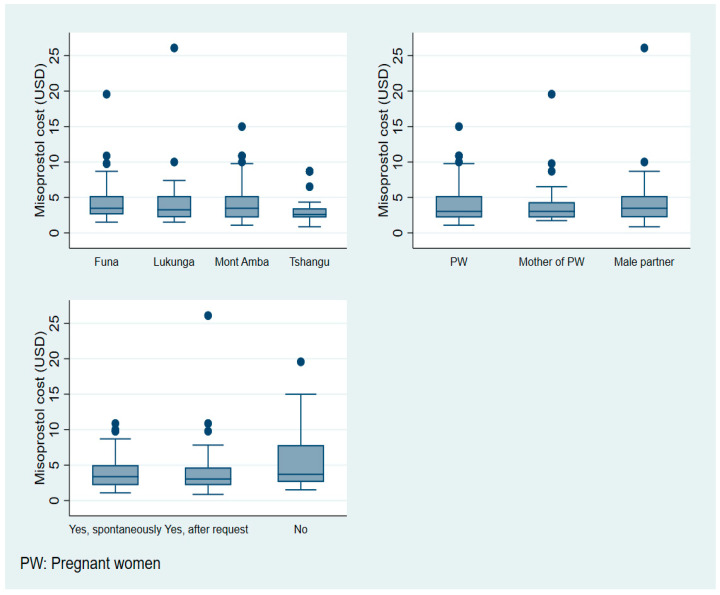
Distribution of misoprostol cost according to district, MC scenario, and whether medical prescription was shown.

**Table 1 healthcare-13-00491-t001:** Outcome of mystery client by district in the city of Kinshasa.

	District									
	Funa		Lukunga		Mont Amba		Tshangu		Overall	
	*n*	%	*n*	%	*n*	%	*n*	%	*n*	%
Availability of misoprostol/mifepristone										
Yes, only misoprostol is in stock	76	42.2	40	44.4	48	32.0	25	41.7	189	39.4
Yes, only mifepristone (Miso 200 or other) is in stock	2	1.1	0	0.0	2	1.3	4	6.7	8	1.7
Yes, misoprostol and Mifepak/combipack or other are in stock	5	2.8	1	1.1	1	0.7	1	1.7	8	1.7
He did not require the prescription but recommended me misoprostol/mifepristone elsewhere	15	8.3	3	3.3	11	7.3	8	13.3	37	7.7
None of these products were in stock	56	31.1	38	42.2	44	29.3	14	23.3	152	31.7
We did not even talk about the medication: misoprostol or mifepristone	23	12.8	7	7.8	37	24.7	5	8.3	72	15.0
Missing	3	1.7	1	1.1	7	4.7	3	5.0	18	3.8
Misoprostol available	81	45.0	40	44.40	48	32.0	25	41.7	194	40.4
What did you get at the pharmacy? *										
Misoprostol [Miso 200 or other]	81	97.6	40	97.6	48	92.3	25	83.3	194	94.2
Mifepak/combipack or other	2	2.4	1	2.4	4	7.7	5	16.7	12	5.8
Total	83	100.0	41	100.0	52	100.0	30	100.0	206	100.0
The scenario										
Pregnant women	120	66.7	30	33.3	90	60.0	30	50.0	270	56.3
Mothers of pregnant women	30	16.7	0	0.0	60	40.0	0	0.0	90	18.8
Male partner	30	16.7	60	66.7	0	0.0	30	50.0	120	25.0
In this Pharmacy, did you show the prescription directly or spontaneously?										
I spontaneously presented the prescription during the scenario	110	61.1	53	58.9	78	52.0	34	56.7	275	57.3
I showed the prescription when the pharmacist/pharmacy attendant asked me.	27	15.0	16	17.8	25	16.7	12	20.0	80	16.7
I completed the scenario without showing the prescription in this pharmacy	43	23.9	21	23.3	45	30.0	14	23.3	123	25.6
Missing	0	0.0	0	0.0	2	1.3	0	0.0	2	0.4
Total	180	100.0	90	100.0	150	100.0	60	100.0	480	100.0

* where misoprostol and/or mifepristone were in stock.

**Table 2 healthcare-13-00491-t002:** Information given by pharmacy providers according to mystery client profiles.

			Pregnant Women		Mothers of Pregnant Women		Male Partners		Total	
			*n* (270)	%	*n* (90)	%	*n* (120)	%	*n* (480)	%
The provider asked:										
Last menstrual period/month of pregnancy			85	31.5	32	35.6	30	25.0	147	30.6
Whether the pregnancy was confirmed *			77	28.5	32	35.6	63	52.5	172	35.8
How the pregnancy was confirmed			70	25.9	15	16.7	34	28.3	119	24.8
Reasons for abortion			124	45.9	41	45.6	64	53.3	229	47.7
Accuracy of information provided:										
Accuracy of dosage instructions *			37	30.1	7	24.1	1	2.4	45	23.2
Accuracy of administration route instructions			74	51.7	23	52.3	24	36.4	121	47.8
Complications that may occur during the abortion process *			19	15.1	0	0.0	19	39.6	38	18.4
At least one accurate possible side effect that may be seen during the abortion			73	59.3	19	65.5	23	54.8	115	59.3
The provider advised:										
What to expect during the abortion process			12	9.6	3	9.7	6	14.0	21	10.6
Where to go in case of complications			30	21.0	1	2.3	32	48.5	63	24.9
To consult the doctor on the fifteenth day to confirm whether the abortion is complete *			46	32.2	3	6.8	25	37.9	74	29.2
The risks to the ongoing pregnancy *			57	39.9	5	11.4	20	30.3	82	32.4

* *p* < 0.001.

## Data Availability

The dataset used for analysis can be obtained upon reasonable request by writing an email to the corresponding author.

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
