# Peer review of "Quality of Information on Medication Abortion in Private Pharmacies: Results from a Mystery Client Study in Kinshasa, Democratic Republic of Congo"

_healthcare, 2025, doi:10.3390/healthcare13050491_

Round 1
Reviewer 1 Report
Comments and Suggestions for Authors
REVIEW
Quality of information on pharmacological abortion in the private setting - Pharmacies: results of a mystery client study in Kinshasa, Democratic Republic of Congo.
The article presents a relevant study on the quality of information provided in pharmacies regarding pharmacological abortion in Kinshasa. The study addresses an important public health issue, with implications for access to safe abortion services in a context of legislative changes.
Areas for Improvement and Recommendations
Despite the quality of the study, there are several points that require revision before publication can be considered:
- In some cases, quantitative data are densely presented in tables and text without sufficient contextualization. It would be advisable to include additional graphs or reformat the presentation of data to improve clarity. Simplifying the tables would help to make them easier to understand.
- Table 1 is not presented in the text, its previous presentation described is suggested. Figure 1 and Figure 2, but their contents are not sufficiently described in the text. Explicit reference to the findings they illustrate is suggested.
- It is mentioned that male partners receive better information than pregnant women. Could this be due to gender bias in pharmaceutical care? Further discussion could be expanded with more sociocultural context.
- In the methodology section, it is mentioned that the sampling included 35 health zones in Kinshasa, but it is not made clear how representativeness and geographic balance were ensured.
- The classification of the quality of the information (score from 0 to 12) is well explained, but it would need to be clarified whether this method has been validated in previous studies.
- Although the study mentions that it had ethical approval, it would be advisable to include a more detailed section on the ethical implications of the use of mystery shopping in the context of pharmacological abortion. In addition, it should be justified why the pharmacists were included in the study without their knowledge or consent. This aspect should be taken into account since the location of the pharmacies in which the study was conducted is provided. This violates the anonymity of the pharmacies studied.
Comments on the Quality of English Language
.
Author Response
Quality of information on pharmacological abortion in the private setting - Pharmacies: results of a mystery client study in Kinshasa, Democratic Republic of Congo.
The article presents a relevant study on the quality of information provided in pharmacies regarding pharmacological abortion in Kinshasa. The study addresses an important public health issue, with implications for access to safe abortion services in the context of legislative changes.
Areas for Improvement and Recommendations
Despite the quality of the study, there are several points that require revision before publication can be considered:
- In some cases, quantitative data are densely presented in tables and text without sufficient contextualization. It would be advisable to include additional graphs or reformat the presentation of data to improve clarity. Simplifying the tables would help to make them easier to understand.
- Table 1 is not presented in the text, its previous presentation described is suggested. Figure 1 and Figure 2, but their contents are not sufficiently described in the text. Explicit reference to the findings they illustrate is suggested.
Thanks for the comment, Table 1 is now cited in the text, which is presenting “Outcome of mystery client by district in the city of Kinshasa”: Table 1 is showing that Misoprostol….
Figure 1 is cited in the text and can be seen as follows: which is a map is the geographic distribution of the private pharmacies surveyed by the mystery clients in Kinshasa shown in Figure 1.
Figure 2 is cited in the text and can be found as follows: A comprehensive examination Figure 2 demonstrated substantial discrepancies in information quality among districts, client scenarios, and prescription presentations [p < 0.001 for all comparisons]
Even Figure 3 is cited in the text: Figure 3 shows the median costs of misoprostol who were comparable across all districts, suggesting general price homogeneity: Funa (3.45 USD), Lukunga (3.26USD), Mont Amba (3.48USD) and Tshangu (2.61USD).
- It is mentioned that male partners receive better information than pregnant women. Could this be due to gender bias in pharmaceutical care? Further discussion could be expanded with more sociocultural context.
Authors answer: We appreciate this question and this fact was discussed in the revised version. The following points may help to understand why male partners were more likely to receive comprehensive information: In many Africans societies, men are traditionally seen as decision-makers and protectors, while women are expected to be more passive and dependent. This dynamic can lead to pharmacists providing more detailed information to men, assuming they are the ones making the final decisions. Women seeking abortion services often face significant stigma and judgment. Pharmacists may provide less information to women to protect their privacy and avoid drawing attention to their situation. There is a perception that men should be involved in reproductive health decisions, especially when it comes to supporting their partners. This can lead to pharmacists giving more comprehensive information to male partners to ensure they are informed and can provide support.
Also, “Gender sensitivities and roles within the community may explain this difference. Due to stigmatization, women may feel uncomfortable staying in a pharmacy for an extended period, fearing suspicion. In contrast, when a man is involved—regardless of whether the pharmacist is a woman—the amount of time spent in the pharmacy receiving information is less likely to be questioned. While men received less correct information on dosage, they received more information on possible complications than our other client profiles; given the important role men can often play in women’s abortion trajectories, this information asymmetry could have important complications for women’s correct use of abortion pills. This could be due to gendered expectations about who will make decisions about seeking follow-up treatment, or to ensure male partners can properly take care of their spouse after the abortion”.
- In the methodology section, it is mentioned that the sampling included 35 health zones in Kinshasa, but it is not made clear how representativeness and geographic balance were ensured.
Authors answer: We appreciate this question. We added the following text to our methods section, line 150-158: “The sample size was determined to accurately evaluate the proportion of adequate information provided by pharmacies in Kinshasa regarding abortion medication. A proportion of 50% was used due to the absence of sufficient data throughout the study's design phase. Utilizing the Kish-Leslie formula for power calculation with a 5% margin of error and a 95% confidence interval, the minimum requisite sample size was determined to be 384. We incorporated a 20% contingency for potential issues such as Potential inaccuracies in pharmacy addresses, refusal of specific pharmacists to serve clients, or the closure of pharmacies during mystery clients visits, resulting in a total of 480 pharmacies being surveyed in this study”
However, the study did not intend to compare outcomes by district(geographic aspect), referred to as region, but rather to gather overall data for the province of Kinshasa. The sampling did not consider that factor, and we performed a random sampling instead.
- The classification of the quality of the information (score from 0 to 12) is well explained, but it would need to be clarified whether this method has been validated in previous studies.
Authors answer: We appreciate this question. The 12 questions that make up the quality score we used in this article come from a similar study
- Nepal “Sigdel A, Angdembe MR, Khanal P, Adhikari N, Maharjan A, Paudel M (2022) Medical abortion drug dispensing practices among private pharmacy workers in Nepal: A mystery client study. PLoS ONE17(11): e0278132. https://doi.org/10.1371/journal.pone.0278132”.
- Akinyemi A, Owolabi OO, Erinfolami T, Stillman M, Bankole A. Quality of information offered to women by drug sellers providing medical abortion in Nigeria: Evidence from providers and their clients. Front Glob Womens Health. 2022 Aug 17;3:899662. doi: 10.3389/fgwh.2022.899662. PMID: 36060610; PMCID: PMC9428275.
We added these reference into the text as well (line 211-213).
We have added the following text to the methods section to explain why pharmacies were classified as providing "adequate" or "inadequate" information at a 75% threshold, lines 217-218 : “we used a cutoff of 75%, which was a modification of the Bloom’s cutoff point”
- Akilimali PZ, Mashinda DK, Lulebo AM, Mafuta EM, Onyamboko MA, Tran NT. The emergence of COVID-19 in the Democratic Republic of Congo: Community knowledge, attitudes, and practices in Kinshasa. PLoS One. 2022 Jun 21;17(6):e0265538. doi: 10.1371/journal.pone.0265538. PMID: 35727797; PMCID: PMC9212135.
- Okello G, Izudi J, Teguzirigwa S, Kakinda A, Van Hal G. Findings of a Cross-Sectional Survey on Knowledge, Attitudes, and Practices about COVID-19 in Uganda: Implications for Public Health Prevention and Control Measures. Biomed Res Int. 2020 Dec 4;2020:5917378. doi: 10.1155/2020/5917378. PMID: 34031643; PMCID: PMC7729389.
- Wahidiyat PA, Yo EC, Wildani MM, Triatmono VR, Yosia M. Cross-sectional study on knowledge, attitude and practice towards thalassaemia among Indonesian youth. BMJ Open. 2021 Dec 3;11(12):e054736. doi: 10.1136/bmjopen-2021-054736. PMID: 34862299; PMCID: PMC8647533.
- Bharathi N, Karthikayan S, Ramakritinan CM. KAP study on dengue epidemiology among paramedical students. Int J Fauna Biol 2015;2:62–4.
- Basheer R, Bhargavi PG, Prakash HP. Knowledge, attitude, and practice of printing press workers towards noise-induced hearing loss. Noise Health 2019;21:62–8. 10.4103/nah.NAH_9_19
- Wan Rozita WM, Yap BW, Veronica S. Knowledge, attitude and practice (KAP) survey on dengue fever in an urban Malay residential area in Kuala Lumpur. Malaysian Journal of Public Health Medicine 2006;6:62–7.
- Although the study mentions that it had ethical approval, it would be advisable to include a more detailed section on the ethical implications of the use of mystery shopping in the context of pharmacological abortion. In addition, it should be justified why the pharmacists were included in the study without their knowledge or consent. This aspect should be taken into account since the location of the pharmacies in which the study was conducted is provided. This violates the anonymity of the pharmacies studied.
The research protocol received ethical approval from the Kinshasa School of Public Health (ESP/CE/04/2023). Given the context of DR Congo where abortion has long been condemned by the penal code, and considering the Congolese culture where abortion is an almost "taboo" subject, this study would evaluate the knowledge and practices of pharmacy providers on medical abortion, which would be subject to information bias if the providers were informed that they were participating in a study. They would change their behavior, and the study would not achieve its objectives or would have biased results. Consistent with mystery client methodology, provider informed consent was waived with ethics committee approval. Regarding the location of the pharmacies, only 40% of visited pharmacies provided Misoprostol (194/480), and the findings are provided in an aggregated format, obscuring which pharmacy provides specific services. Additionally, we altered the GPS coordinates to prevent the identification of the precise locations of the pharmacies, Counseling timestamps to prevent the identification of the pharmacy through travel data. Aside from the modified GPS data intended to maintain the pharmacy's privacy, we have neither gathered nor disseminated any more information that could reveal the pharmacy's identity.
Reviewer 2 Report
Comments and Suggestions for Authors
An important study on the structuring of pharmacies within health services.
It would be appropriate to explain the answers to a few questions in the article.
Based on which criteria was the sample size decided?
What are Type I, II and power?
How was hospitality made regarding pharmacy visits by region? Is it in the form of cluster sampling?
Table 1 and Table 2 seem to be Chi-square analysis with the analysis, which test was used should be explained under the table.
It is seen that there are many 1 values ​​in the tables. It should be reviewed again regarding whether the use of Chi-square is appropriate.
Author Response
Reviewer #2
Comments and Suggestions
An important study on the structuring of pharmacies within health services.
It would be appropriate to explain the answers to a few questions in the article.
Authors answer: We thank the reviewer for their detailed constructive feedback and respond to individual comments below.
Based on which criteria was the sample size decided?
What are Type I, II and power?
Authors answer: we added the text in “The sample size was determined to accurately evaluate the proportion of adequate information provided by pharmacies in Kinshasa regarding abortion medication. A proportion of 50% was used due to the absence of sufficient data throughout the study's design phase. Utilizing the Kish-Leslie formula for power calculation with a 5% margin of error and a 95% confidence interval, the minimum requisite sample size was determined to be 384. We incorporated a 20% contingency for potential issues such as Potential inaccuracies in pharmacy addresses, refusal of specific pharmacists to serve clients, or the closure of pharmacies during mystery clients visits, resulting in a total of 480 pharmacies being surveyed in this study” (line 151-159)
How was hospitality made regarding pharmacy visits by region? Is it in the form of cluster sampling?
The study did not intend to compare outcomes by district, referred to as region, but rather to gather overall data for the province of Kinshasa. The sampling did not consider that factor, and we performed a random sampling instead. Thank you for your relevant comments.
Table 1 and Table 2 seem to be Chi-square analysis with the analysis, which test was used should be explained under the table.
It is seen that there are many 1 values ​​in the tables. It should be reviewed again regarding whether the use of Chi-square is appropriate
Authors answer: We have removed the p-value column, as this comparison is not our objective and as noted by the reviewer. The Chi-square test is inappropriate for certain data, notwithstanding its non-application. We were comparing the proportions between groups.
Reviewer 3 Report
Comments and Suggestions for Authors
Dear Authors,
Thank you for submitting your manuscript, which examines the availability of misoprostol and the quality of pharmacy-provided information on medication abortion in Kinshasa. This is a timely and relevant study, particularly given the evolving policy landscape around abortion care. The use of the mystery client methodology provides a valuable lens for assessing pharmacy practices and their implications for reproductive health access. However, several areas require revision and further elaboration to enhance the clarity, depth, and impact of the manuscript. Below are my detailed comments and suggestions.
Title and Abstract
The title accurately reflects the study's scope, but it could be more explicit about the mystery client methodology to improve discoverability. The abstract effectively summarizes key findings but would benefit from greater specificity regarding:
- The total number of pharmacies visited and the proportion that provided misoprostol.
- A brief mention of ethical considerations regarding the mystery client approach.
- A stronger concluding statement linking findings to policy or practice recommendations.
Introduction
The introduction establishes the importance of pharmacy-based abortion services but could be more structured to ensure clarity. Consider the following improvements:
- Provide a clearer background on why pharmacies play a critical role in abortion access in Kinshasa.
- Expand on existing literature regarding mystery client studies on medication abortion in similar contexts, such as those in Nigeria, India, and Nepal.
- Clarify the study’s specific contribution. How does it build upon prior research? Does it explore novel aspects of information quality or pharmacy-based abortion access?
Methods
The methodology is robust, but additional details would improve transparency and reproducibility:
- Provide more information on how mystery clients were trained to ensure consistency in responses. Did they receive scripts or standardized training modules?
- Justify the choice of a 12-point information quality score. Was this adapted from existing frameworks, or was it developed specifically for this study?
- Explain why pharmacies were classified as providing "adequate" or "inadequate" information at a 75% threshold and whether alternative cutoffs were considered.
- Detail how data were collected and validated, particularly given the reliance on mystery clients' recall. Were there mechanisms in place to minimize recall bias?
Results
The results are well-structured and informative, but they could be made clearer with additional interpretation:
- Provide a clearer summary of key statistics at the beginning of each section to help the reader navigate the data.
- Quantify the extent of missing data, if applicable, and discuss any potential biases introduced by missing responses.
- Expand on the differences in provider responses based on client profiles. Why were male partners more likely to receive comprehensive information? What cultural or social norms might explain this disparity?
Discussion
The discussion is comprehensive but would benefit from deeper engagement with the following issues:
- Comparison with existing literature: The study references prior research, but a more detailed comparison with studies in similar contexts (e.g., Kenya, Senegal, Nigeria) would strengthen the argument.
- Explanations for observed disparities: The study finds that pharmacy providers offered different levels of information based on client identity. Could gender biases, stigma, or provider assumptions about client knowledge play a role?
- Geographic disparities: The variation in information quality across districts is significant. Discuss potential systemic factors (e.g., regulatory oversight, socioeconomic disparities, or training programs) that may explain these differences.
- Public health implications: How can pharmacy-based abortion counseling be improved? Consider policy recommendations, such as standardized training programs for pharmacy staff or increased regulatory oversight of medication abortion dispensing.
Limitations and Future Directions
The limitations section is well-articulated, but additional details could further strengthen it:
- Discuss the potential influence of observer bias, as mystery clients may interpret interactions differently.
- Consider whether self-selection bias could impact results, given that pharmacies stocking misoprostol may differ systematically from those that do not.
- Future research directions could include longitudinal studies assessing how pharmacy information quality evolves following policy changes or training interventions.
Conclusion
The conclusion effectively summarizes key findings but could be more action-oriented.
- Provide clearer recommendations on how findings should inform policy, training, and future research.
- Highlight the practical implications for public health, pharmacy regulation, and reproductive rights advocacy in Kinshasa.
